# Differences in Cortical Area Activity and Motor Imagery Vivid-Ness during Evaluation of Motor Imagery Tasks in Right and Left Hemiplegics

**DOI:** 10.3390/brainsci13050748

**Published:** 2023-04-29

**Authors:** Kengo Fujiwara, Masatomo Shibata, Yoshinaga Awano, Naoki Iso, Koji Shibayama, Toshio Higashi

**Affiliations:** 1Medical Corporation Zeshinkai Nagasaki Rehabilitation Hospital, Nagasaki 850-0854, Japan; m-shibata@zeshinkai.or.jp (M.S.); k-shiba-3@zeshinkai.or.jp (K.S.); 2Graduate School of Biomedical Sciences, Nagasaki University, Nagasaki 852-8520, Japan; higashi-t@nagasaki-u.ac.jp; 3School Corporation Tamaki Gakuen Nagasaki College of Medical Technology, Nagasaki 850-0822, Japan; awano@zeshinkai.or.jp; 4Faculty of Health Sciences, Tokyo Kasei University, Saitama 350-1398, Japan; iso-n@tokyo-kasei.ac.jp

**Keywords:** stroke, motor imagery vividness, NIRS

## Abstract

The ability to develop vivid motor imagery (MI) is important for effective mental practice. Therefore, we aimed to determine differences in the MI clarity and cortical area activity between patients with right hemiplegia and left hemiplegia after stroke in an MI task. In total, 11 participants with right hemiplegia and 14 with left hemiplegia were categorized into two groups. The MI task required the flexion and extension of the finger on the paralyzed side. Considering that MI vividness changes with MI practice, we measured the MI vividness and cortical area activity during the task before and after MI practice. MI vividness was evaluated subjectively using the visual analog scale, and cerebral hemodynamics during the task were measured using near-infrared spectroscopy in cortical regions during the MI task. The MI sharpness and cortical area activity in the MI task were significantly lower in the right hemiplegia group than in the left hemiplegia group. Therefore, when practicing mental practices with right hemiplegia, it is necessary to devise ways by which to increase MI vividness.

## 1. Introduction

There are many rehabilitation methods used for motor paralysis after stroke, including motor therapy, electrical stimulation therapy, constraint-induced movement therapy, action observation (AO), and mental practices (MP) [1,2,3,4]. Among these, MP is the continuous repetition of motor imagery (MI) to improve the performance of motor tasks and can be performed regardless of the degree of motor paralysis or the physical environment.

A systematic review of MP in individuals with stroke reported its interventional effects on post-stroke motor paralysis [5]. Behind this intervention effect, it has been established that MI induces activity in the cortical regions that is comparable to that during actual movement [6]. The combination of the brain–computer interface, MP, and occupational therapy has been reported as a promising approach by which to promote sensory-motor recovery and the functional independence of the upper limb in the daily activities of individuals with stroke [7]. On the other hand, it is important for the effective practice of MP to know how vividly the individual can recall the MI of the task, as some studies have reported that individuals with stroke have difficulty recalling the MI of the task [8]. The regions and degrees of brain activity differ between the first-person and third-person motor imagery [9]. However, during the first-person motor imagery of the upper extremities, the premotor cortex, auxiliary motor, primary motor, primary somatosensory, superior parietal lobe, and inferior parietal lobe are activated, similar to the brain activity observed during actual movement [10].

As a neurophysiological study on the vividness of MI, by performing MI while grasping the object used in the task in the same posture as the task, proprioceptive information is provided during MI and the excitability of the corticospinal tract is enhanced. Increasingly, this has been reported to affect MI [11,12]. It has also been shown that performing MI while viewing a video of a task, which is called action observation (AO), increases MI vividness [13]. This AO+MI intervention also increases activity in motor-related areas, such as the supplementary motor cortex and the primary motor cortex, more than AO and MI alone [14]. There is evidence that the independent use of MI and AO is largely effective, and that the two processes can induce similar changes in motor system activity [15]. In a study that evaluated the effects of a video presentation on AO, the corticospinal tract was more excited when presented with a first-person perspective (observing from one’s own perspective) than with a third-person perspective (observing from another person’s perspective) [16].

In stroke patients, it is necessary to practice MP according to the paralyzed side. However, it is questionable as to whether the MP of patients with right hemiplegia and left hemiplegia can be combined. Among them, there is a report describing the difference in the laterality of the cerebral hemispheres. Regarding the differences in cerebral hemispheric function during MI tasks, a study using transcranial magnetic stimulation in healthy individuals reported increased excitability in the primary motor cortex of the left hemisphere in left, right, and bilateral MI, indicating that the dominant hemisphere in MI is the left hemisphere [17]. Studies of individuals with stroke have also stated that the predominant hemisphere for MI is the left hemisphere [18]. These studies have shown that MI is predominantly in the left hemisphere; however, no studies have compared MI clarity and cortical area activity in patients with right and left hemiplegia after stroke.

In this study, we compared MI vividness and cortical area activity in individuals with stroke and with right and left hemiplegia after stroke, and clarified the differences between individuals with stroke and with right and left hemiplegia in MI tasks.

## 2. Materials and Methods

### 2.1. Participants

The participants included 25 individuals with cerebral infarction (mean age: 68.1 ± 11.0 years) admitted to a recovery rehabilitation ward (Table 1). The mean time from onset was 26.1 ± 7.9 days.

On the paraplegic side, 11 individuals with stroke were in the right hemiplegia group (mean age: 71.6 ± 9.5 years) and 14 were in the left hemiplegia group (mean age: 65.4 ± 11.7 years). The dominant hand was determined by self-report, and all participants were right-handed without any experience in correcting their dominant hand. The participants’ upper limb function on the paralyzed side scored 42.4 ± 20.3 points on the Fugl–Meyer Assessment (FMA), 1.7 ±1.6 points on the Motor Activity Log (MAL) Amount of Use (AOU), and 1.6 ± 1.5 points on the Quality of Movement (QOM). The Fugl–Meyer Assessment (FMA) is a measure of motor paralysis function in the upper extremity. The Motor Activity Log (MAL) Amount of Use (AOU) points on the Quality of Movement (QOM) are quantity and quality measures, respectively, of how well the paralyzed upper extremity is used in daily activities. The vividness of motor imagery was VAS1 before MI practice and VAS2 after MI practice. VAS1 was 64.6 ± 27.3 and VAS2 was 72.7 ± 25.2.

The exclusion criteria were as follows: (1) history of neurological disease, respiratory disease, stroke, or dementia; (2) a score of 23 or less on the Mini Mental State Examination (MMSE) of cognitive function; (3) individuals with stroke who were unable to perform the experimental tasks due to impaired consciousness, aphasia, hemiplegia, body apraxia, or visual or spatial perception effects; (4) inability to sit in a backrest chair or wheelchair; and (5) blood pressure fluctuations caused by maintaining a sitting posture.

This study was approved by the Ethics Committee of the Department of Health Sciences, Nagasaki University Graduate School of Biomedical Sciences (Nagasaki, Japan; approval number: 15070927). The participants were given a full explanation of the research and were asked to sign a consent form before participating.

### 2.2. MI Task

The MI task consisted of the flexion and extension of the paralyzed hand [19]. The flexion and extension movements consisted of a series of movements in which the hand was held open, the palm was closed for 1 s, and opened for the next 1 s [20]. The participants were instructed to perform myosensory (first-person) MI as if they were performing MI themselves and not to move their hands (no muscle contraction) during the MI [20,21]. Video images were used to explain the MI tasks to the participants. Video images were captured using a digital camera (D5100, Nikon Corporation) from the first-person viewpoint of a healthy adult individual in left and right hand flexion and extension (Figure 1). Therefore, the video presented during the MI task performed in this experiment was a first-person viewpoint video of AO + MI of another person’s hand. Video images were taken from the first-person perspective of the flexion and extension of the left and right fingers of a healthy adult and were used to explain the MI task to the participants and the use of video images during MI practice.

### 2.3. Experimental Procedure

The participants were shown a video image explaining the MI task, and the experiment was started after confirming that the participant’s understanding. The experiments were conducted within 1 week of admission. Considering that MI vividness changes with MI practice, we measured the MI vividness and cortical area activity during the MI tasks before and after MI practice. MI vividness was subjectively assessed using a visual analog scale (VAS) [20], and cerebral hemodynamics during the MI task were measured using near-infrared spectroscopy (NIRS) for activity in cortical regions during the MI task. Table 1 shows VAS1 for MI vividness before MI practice and VAS2 for MI vividness after MI practice. For the MI exercise, the participants practiced MI for 5 min while being shown a video image. The participants were instructed to observe the paralyzed upper limb during the NIRS measurements to facilitate MI.

### 2.4. NIRS Measurement and Analysis

NIRS measurements were performed using an optical topography system (ETG4000; Hitachi Medical Corporation). The participants were seated in a chair or wheelchair position, with both upper limbs resting on a table. NIRS probes were arranged in a 4x4 optode probe configuration for Cz according to the international 10–20 method. With a total of 24 channels, the distance between the optodes was 3.0 cm [22] (Figure 2). The NIRS system emitted at two different wavelengths (625 and 830 nm) over the scalp, and the relative change in the absorption of the near-infrared light was measured. These values were based on the modified Beer–Lambert [23,24,25], oxygenated hemoglobin (oxy-Hb), and deoxygenated hemoglobin concentrations. It has been reported that there is no difference in the optical path length of the left and right target regions for this NIRS probe [23].

The regions of interest (ROIs) included the left and right sensorimotor cortex (SMC), premotor area (PMA), prefrontal cortex (PFC), supplementary motor area (SMA), and anterior SMA (pre-SMA) (Figure 2). The 15 s data from 5 s after the start of the task to 20 s after the end of the task were used as the data during the task.

Based on previous studies, we designated 18 and 22ch as Left-SMC; 21 and 24ch as Right-SMC; 9, 12, 13, and 16ch as SMA; 2, 5, and 6ch as Pre-SMA; 8, 11, and 15ch as Left-PMA; 10, 14, and 17ch as Right-PMA; 1 and 4ch as Left-PFC; 3 and 7ch as Right-PFC; Left-PFC for channels 1 and 4; and Right-PFC for channels 3 and 7 [21,26,27]. For this SMC, the sensory and motor cortices were treated as in the NIRS studies [21,27].

NIRS measurements were performed in a block design with three consecutive cycles of alternating 20 s MI tasks and 30 s rest periods [28,29]. Because oxy-Hb is a more obvious indicator of activation than deoxy-Hb [30,31], the amount of change in oxy-Hb during the MI task was used as an indicator of regional cerebral hemodynamics. The data obtained were analyzed for changes in the oxy-Hb concentration in each region in the integral mode, which was calculated by adding up and averaging the data from three cycles. The data used during the task were considered for the time it takes for cerebral blood flow to increase with neural activity, and the 15 s data from 5 s after the start of the task to 20 s after the end of the task were used as the data during the task [28] (Figure 3). The measured data were filtered with a 3 Hz high-pass filter on 0.1 standard deviations of wave analysis, as used by previous researchers [28,32]. This filter was used to remove noise, such as hyperactivity due to skin and blood dynamics and marked channels with high noise levels [33]. If obvious artifacts were detected, they were removed from the waveforms. The average waveform was calculated using integral analysis. The oxy-Hb values for each region were converted into Z-scores.

### 2.5. Assessment of Paralyzed Upper Extremity Function

Upper extremity function was assessed using the Fugl–Meyer Assessment (FMA), Motor Activity Log (MAL), Amount of Use (AOU), Quality of Movement (QOM), and Motor Activity Log within 1 week of admission.

### 2.6. Statistics

Statistical analysis software (SPSS v22.0; IBM Corp., Armonk, NY, USA) was used to compare the oxy-Hb and VAS of ROIs (PFC, PMA, pre-SMA, SMA, and SMC) during MI before and after MI practice in the right and left hemiplegia groups. The Wilcoxon signed-rank test was used to compare the FMA and MAL between the right and left hemiplegia groups. All significance levels were set at less than 5%.

## 3. Results

There were no significant differences in the upper limb function of the paralyzed side between the right hemiplegic group (FMA 46.5±5.3, MAL [AOU] 2.2±0.5, and MAL [QOM] 2.0±0.5) and the left hemiplegic group (FMA 39.1±6.0, MAL [AOU] 1.3±0.4, and MAL [QOM] 1.2±0.4) (Figure 4).

There were no significant differences in the FMA and MAL (AOU and QOM) scores between the right and left hemiplegic groups in terms of paralyzed upper limb function.

Regarding MI vividness, the VAS was evaluated before and after the MI practice. No significant difference was found in the right hemiplegia group before MI practice (65.6 ± 9.3) and after MI practice (68.7 ± 9.0). For the left hemiplegia group, there was a significant increase in the VAS before MI practice (63.9 ± 6.8) and after MI practice (75.8 ± 5.7) (*p* < 0.05) (Figure 5).

Regarding the cortical area activity during the MI task, the right hemiplegic group showed lower oxy-Hb levels during MI in all ROIs than the left hemiplegic group. The left hemiplegia group showed significantly higher values for the left and right PFC (*p* < 0.05) (Figure 6).

Activity in the cortical regions during the MI task showed lower oxy-Hb levels during MI in all ROIs in the right hemiplegia group than in the left hemiplegia group, and significantly higher values for the left and right PFC in the left hemiplegia group.

## 4. Discussion

The purpose of this study was to compare MI vividness and cortical area activity in right and left hemiplegia groups after stroke and to determine the differences between the right and left hemiplegia groups in MI tasks.

Regarding MI vividness, there was no significant difference in the right hemiplegic group before and after MI practice; however, there was a significant increase in the left hemiplegic group. In addition, in terms of cortical area activity during the MI task, oxy-Hb during MI was lower in the right hemiplegic group than in the left hemiplegic group in all ROIs. This is a difference in the cerebral hemispheric function during MI. Studies on healthy people and individuals with stroke have reported that the predominant hemisphere in MI is the left hemisphere [28,29]. Regarding the relationship between MI vividness and cortical area activity, a study on MI vividness and corticospinal tract excitability in MI tasks in healthy individuals showed a positive correlation between MI vividness and corticospinal tract excitability [34]. These results suggest that the dominant hemisphere for MI is the left hemisphere, making MI recall more difficult in the right hemiplegic group with left hemisphere damage. In addition, activity in the cortical areas did not increase due to low MI vividness after MI practice.

One study reported that the PFC area increased during MI. In a NIRS study using a smartphone flick input task, the left SMC, SMA, and right PFC showed significant changes in cerebral hemodynamics as the task cycle progressed, demonstrating that movement was learned over time. These activity changes may reflect different aspects of motor task acquisition, such as increased finger motor activity, motor inhibition, and visual working memory, in the left SMC, SMA, and right PFC, respectively [27]. Therefore, it is possible that the activity of the right PFC region was higher than that of the right SMC and SMA in the present experiment in the early stage of learning in the MI task.

Regarding the method of increasing MI vividness, previous neurophysiological investigations have reported that performing MI in the same posture as the task [11] and grasping the object used for the task affects the intrinsic sensory information acquired in the MI [12] and increases the excitability of the corticospinal tract during MI. Research that combines MI and AO as a means of enhancing the presence of MI is attracting attention, and research is also being conducted on video presentations. Studies using such video presentations have shown differences in the perspectives used in AO, with first-person perspectives being more effective than third-person perspectives in inducing activity in the brain’s somatosensory cortex [16]. In addition, the results of one study showed that PMA brain activity changes less when the image of another person’s hand is displayed compared to when the image of one’s own hand is displayed [35]. A study examining the effect of the hand angle on the presented image showed that images with the third finger rotated to the midline direction of the body responded faster than images rotated in two opposite directions (medial-lateral). Effectiveness was observed and it was speculated that the MI strategy was predominantly employed [36,37]. First-person perspective (kinesthetic MI) video presentation is, thus, ideal, and it is clear that the subject’s own hand should be shown instead of someone else’s. Hands in biomechanically constrained positions are also thought to be susceptible to MI.

Performing MI while watching a video of a task, as in the present case, enhances MI vividness by combining the AO and MI [13]. In this study, MI practice was conducted while the participants watched a 5 min video of the model’s hands moving. However, in the right hemiplegic group, 5 min of MI practice did not improve either MI vividness or cortical area activity. A study conducted in individuals with stroke patients and healthy controls that used inverted images of their own hands to enhance the vividness of MI and cortical area activity revealed that the vividness of MI and cortical area activity were higher when images of one’s own hands were used, compared to when images of other people’s hands were used [38,39]. In a study evaluating the task performance of AO + MI, dart throwing, basketball throwing, and golf tasks were performed under the conditions of AO only and MI only; AO + MI was shown to be superior to MI alone [40,41,42,43]. From these studies, it can be concluded that the conditions for better performance are the use of videos of participants’ own hand and the combination of AO + MI.

Therefore, when performing MP on individuals with stroke and with right hemiplegia in clinical practice, it is necessary to consider using inverted images of individuals’ own hand as a method of enhancing MI vividness. In addition, although the MI task in this study was the flexion and extension of the paralyzed side of the hand, it has been reported that younger individuals with stroke have more heterogeneous needs than older individuals with stroke and require a more individualized program [44]. Therefore, it is necessary to consider participant preferences in clinical practice.

## 5. Limitations

One of the limitations of the study is the small number of participants; therefore, an analysis that considers the site of injury could not be performed. In addition, only cerebral hemorrhage and cerebral infarction were included in this study due to the small sample size. A further analysis that considers disease and uses a larger sample size is required. NIRS was used to analyze the activity of cortical regions, and mainly motor-related regions. However, measuring the activity of other cortical regions was limited because other neuropsychological measures were not used.

## 6. Conclusions

MI practice was performed while showing AO + MI images of other people’s hands to patients with right and left hemiplegia due to stroke. Although there was no significant difference in the upper extremity function of the affected side between groups, the MI vividness and cortical area activity during the MI task were lower in the right hemiplegia group than in the left hemiplegia group. Therefore, when practicing MP for a right hemiplegic patient, it is necessary to improve the clarity of MI, such as using a reversed image of the patient’s own hand to facilitate MI.

## Figures and Tables

**Figure 1 brainsci-13-00748-f001:**
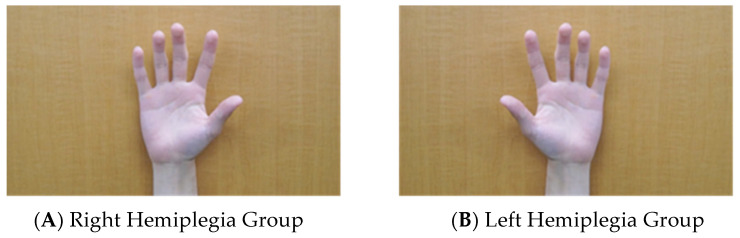
First-person model video of left and right hand flexion and extension movements of a healthy adult. The video presented during the MI task performed in this experiment was a first-person viewpoint video of AO + MI of another person’s hand. For the right hemiplegia, (**A**) the AO + MI image of another person’s hand, and for the left hemiplegia, (**B**) the AO + MI image of another person’s hand, were used for MI practice. Participants practiced MI for 5 min while being shown a video image. The participants were instructed to observe the paralyzed upper limb during the NIRS measurements to facilitate MI.

**Figure 2 brainsci-13-00748-f002:**
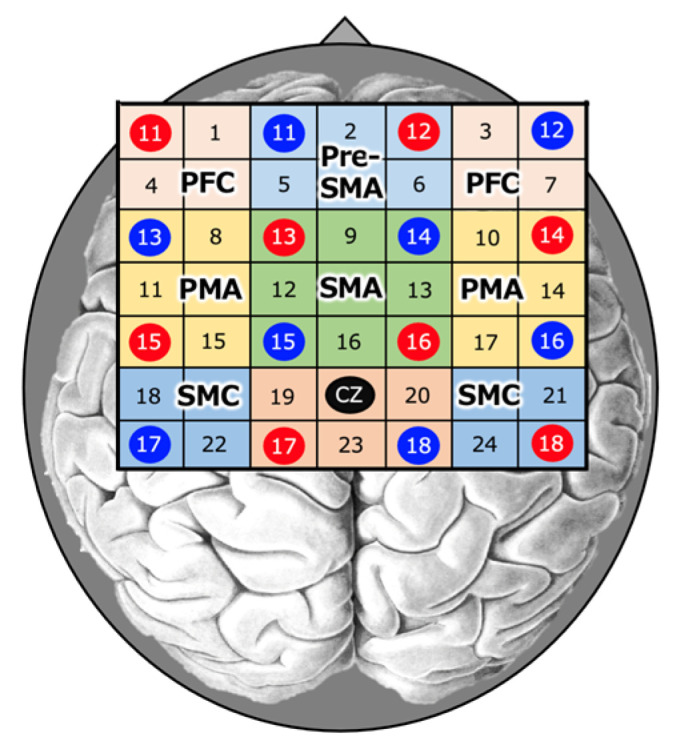
Near-infrared spectroscopy (NIRS). NIRS measurements were performed using an optical topography system. NIRS probes were positioned at the Cz position (midpoint of the crown of the head) according to the international 10–20 method in a 4 × 4 probe set.

**Figure 3 brainsci-13-00748-f003:**
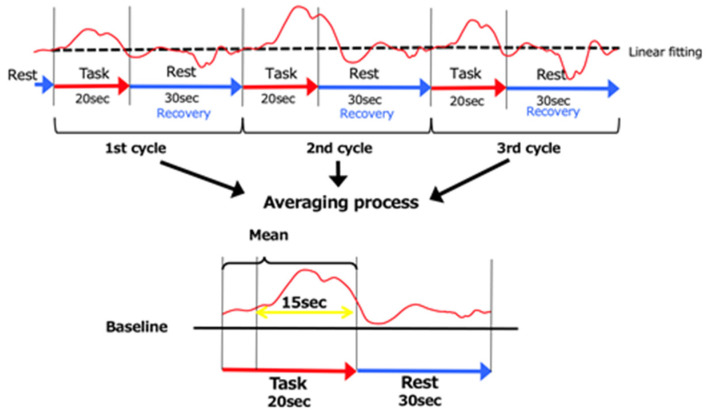
The 3 cycles of data. As for the baseline, the data were averaged over the 5 s immediately prior to the start of the MI task and the 5 s following its completion. The data recorded 5 s following the start of the MI task until 20 s following its completion (15 s of data) were used.

**Figure 4 brainsci-13-00748-f004:**
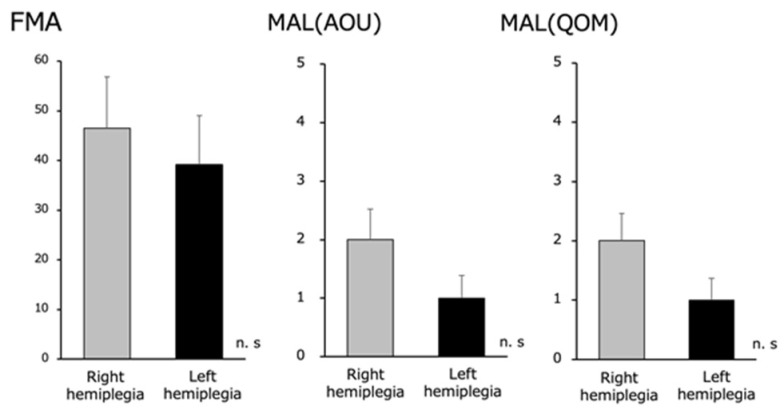
Fugl–Meyer Assessment (FMA) and Motor Activity Log (MAL) (Amount of Use [AOU] and Quality of Movement [QOM]) in the right and left hemiplegia groups. We found no difference in the upper extremity function between the right and left hemiplegia.

**Figure 5 brainsci-13-00748-f005:**
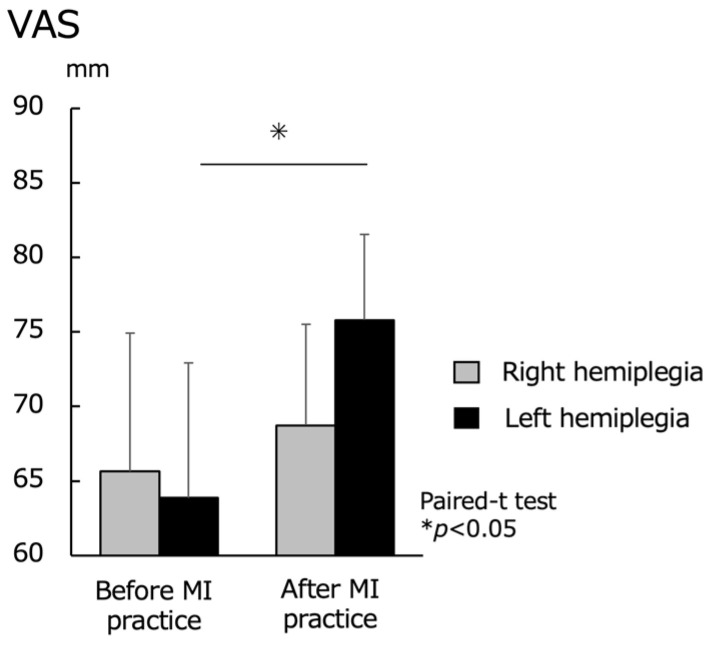
Motor imagery (MI) vividness before and after MI practice for the right hemiplegia and left hemiplegia groups. VAS, visual analog scale. There was no significant difference in the upper extremity function between the right and left hemiplegia, but there was a significant difference in the MI vividness after MI practice.

**Figure 6 brainsci-13-00748-f006:**
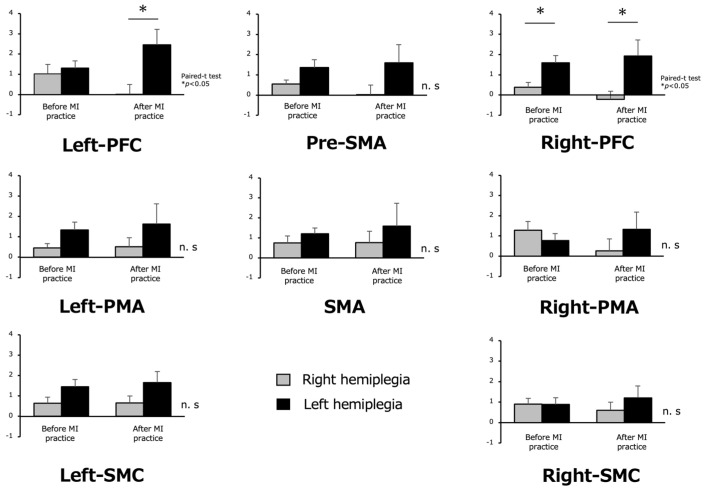
Oxygenated hemoglobin during motor imagery (MI) before and after MI practice in the right hemiplegia and left hemiplegia groups. In the right and left hemiplegia, the oxy-Hb values in the left hemiplegia were high in all regions, and a significant difference was observed in left and right PFC. The other results were not significant (n.s) different.

**Table 1 brainsci-13-00748-t001:** The basic attributes of the study participants.

ID	Sex	Age	Days since Onset	Region of Damage	Stroke Side	MMSE	FMA	MAL (AOU)	MAL (QOM)	VAS1	VAS2
1	M	78	20	Left anterior cephalic lobe, lateral lobe subcortical	R	25	40	0.8	0.7	50	56
2	M	57	54	Left anterior cephalic lobe	R	28	42	0.4	0.4	48	60
3	M	87	22	Left paraventricular white matter	R	29	49	2.8	2.2	100	100
4	F	86	33	Left side of the pons	R	24	63	5.0	4.4	100	100
5	M	72	24	Left thalamus	R	28	51	1.2	1.3	100	97
6	M	68	31	Left intension	R	24	65	4.6	3.9	21	24
7	M	67	18	Left crown of radiation	R	29	56	2.9	2.6	21	23
8	M	68	22	Left intension	R	30	62	3.5	3.8	60	75
9	M	67	30	Left intension	R	28	4	0.0	0.0	52	41
10	F	61	35	Pons	R	26	50	2.0	2.0	100	100
11	F	77	18	Left thalamus	R	29	30	0.6	0.6	70	80
12	F	78	21	Right internal hind leg	L	27	61	1.5	1.5	25	50
13	F	70	23	Right lentiform nucleus	L	30	48	1.4	1.0	100	100
14	M	84	35	Right anterior cephalic lobe	L	24	57	3.0	3.5	28	45
15	M	65	31	Right internal hind leg	L	30	55	0.7	1.2	51	52
16	F	61	25	Right putamen	L	24	23	0.0	0.1	54	63
17	M	38	24	Right temporal lobe	L	28	14	0.0	0.0	100	100
18	F	63	19	Right internal hind leg	L	28	30	2.2	1.7	69	71
19	M	63	21	Right crown of radiation	L	28	60	0.6	0.5	52	66
20	M	60	22	Right internal hind leg	L	30	5	0.0	0.0	100	100
21	M	64	23	Right anterior cephalic lobe	L	24	64	5.0	4.5	53	74
22	M	67	33	Right thalamus	L	24	60	2.3	1.8	48	53
23	M	67	25	Right crown of radiation	L	30	48	1.1	1.3	86	97
24	M	82	19	Right lentiform nucleus	L	28	4	0.0	0.0	78	100
25	M	53	24	Right internal hind leg	L	30	19	0.0	0.0	50	90
AVG		68.1	26.1			27.4	42.4	1.7	1.6	64.6	72.7
SD		11.0	7.9			2.3	20.3	1.6	1.5	27.3	25.2
SE		2.2	1.6			0.5	4.1	0.3	0.3	5.5	5.0

M: Male F: Female R: Right L: Left.

## Data Availability

Not applicable.

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
