# Peer review of "Differences in Cortical Area Activity and Motor Imagery Vivid-Ness during Evaluation of Motor Imagery Tasks in Right and Left Hemiplegics"

_brainsci, 2023, doi:10.3390/brainsci13050748_

Round 1

Reviewer 1 Report

The article entitled “Differences in cortical area activity and motor imagery vividness during evaluation of motor imagery tasks in right and left hemiplegics” presented by Kengo Fujiwara et al. aims to clarify MI and cortical area activity in patients with right and left hemiplegia after stroke. As a result, they found MI sharpness and cortical area activity in the MI task were significantly lower in the right hemiplegia group than in the left hemiplegia group, indicating it is necessary to devise ways to increase MI vividness when practicing MP with right hemiplegia. Overall, the work is of interest although I do have a few concerns and suggestions that I mention below.

1. It is not completely clear how to measure the cortical area activity in the work, the authors should clarify a bit more.

2. The patents are grouped based on the right and left hemiplegics, the legion size and location in the brain are not described at all, which would probably affect the results.

3. The group size is too small, only 25 patients in total were collected.

Author Response

RESPONSES TO REVIEWER 1’S COMMENTS

The authors would like to thank the reviewer for their constructive critique aimed at improving the manuscript. We have made every effort to address the issues raised and to respond to all comments. The revisions are indicated in red font in the revised manuscript. Below is a detailed, point-by-point response to the reviewer’s comments. We hope that our revisions meet the reviewer’s expectations.

Reviewer 2 Report

Dear Authors, 

Thank you for the opportunity to revise your manuscript (ms). Overall, this is a very interesting ms. Please find below some issues that I found while reading your work.

1.     Reorganize the ms, especially the methods and results based on the STROBE guidelines. 

2.     Please include recent literature for the using Mental Practices (MP) in stroke individuals. Indeed ref [5] is outdated. Add for example  Zanona AF, et al. Brain-computer interface combined with mental practice and occupational therapy enhances upper limb motor recovery, activities of daily living, and participation in subacute stroke. Front Neurol. 2023 Jan 9;13:1041978. doi: 10.3389/fneur.2022.1041978. 

3.     Throughout the ms make sure to avoid using the word "patient" in reference to individuals with stroke. Instead, promote person-centered language, i.e., individuals with stroke.

4.     In the Methods, please define with better accuracy the stroke phase.

5.     The individuals that had a stroke in the dominant hemisphere may show speech and communication while the participants with a stroke in the non-dominant hemisphere may report Visual and/or spatial perception impairments. This can influence the study results. How was it accounted for?  

6.     Add the location of the injury

7.     In the results consider to adda figure about the brain areas and the Oxy-Hb during MI before and after MI practice of one right and left right hemiplegia individuals 

8.     In the Discussion section, It would be interesting to consider and discuss the stroke individual perspective during the treatment, for example: Perin C, et al. Differences in Rehabilitation Needs after Stroke: A Similarity Analysis on the ICF Core Set for Stroke. Int J Environ Res Public Health. 2020 Jun 16;17(12):4291. doi: 10.3390/ijerph17124291. 

N/A

Author Response

AUTHORS’ RESPONSES TO REVIEWER 2’S COMMENTS

The authors would like to thank the reviewer for their constructive critique aimed at improving the manuscript. We have made every effort to address the issues raised and to respond to all comments. The revisions are indicated in red font in the revised manuscript. Below is a detailed, point-by-point response to the reviewer’s comments. We hope that our revisions meet the reviewer’s expectations.

Round 2

Reviewer 2 Report

Thank you, 

All comments have been addressed.